# Influence of SARS-CoV2 Pandemic on Colorectal Cancer Diagnosis, Presentation, and Surgical Management in a Tertiary Center: A Retrospective Study

**DOI:** 10.3390/diagnostics15020129

**Published:** 2025-01-08

**Authors:** Roman Taulean, Roxana Zaharie, Dan Valean, Lia Usatiuc, Mohammad Dib, Emil Moiș, Calin Popa, Andra Ciocan, Alin Fetti, Nadim Al-Hajjar, Florin Zaharie

**Affiliations:** 1Regional Institute of Gastroenterology and Hepatology “Octavian Fodor”, 400162 Cluj-Napoca, Romania; axiromplus@gmail.com (R.T.); valean.d92@gmail.com (D.V.); mdibevan@yahoo.ro (M.D.); drmoisemil@gmail.com (E.M.); calinp2003@yahoo.com (C.P.); andra.ciocan10@gmail.com (A.C.); confetti_ro@yahoo.com (A.F.); na_hajjar@yahoo.com (N.A.-H.); florinzaharie@yahoo.com (F.Z.); 2Department of General Surgery, University of Medicine and Pharmacy “Iuliu Hațieganu”, 400347 Cluj-Napoca, Romania; 3Department of Gastroenterology, University of Medicine and Pharmacy “Iuliu Hațieganu”, 400347 Cluj-Napoca, Romania; 4Department of Patophysiology, University of Medicine and Pharmacy “Iuliu Hațieganu”, 400347 Cluj-Napoca, Romania; lia.usatiuc@umfcluj.ro

**Keywords:** COVID-19 pandemic, colorectal surgery, diagnosis, prognosis, morbidity and mortality

## Abstract

**Background**: Oncological surgery during the COVID-19 pandemic was performed only in carefully selected cases, due to variation in the allocation of resources. The purpose of this study was to highlight the impact of the pandemic lockdown on the presentation, diagnosis, and surgical management of colorectal cancers as well as the post-pandemic changes in this area. **Material and methods**: This single center, retrospective comparative study contained 1687 patients, divided into three groups with equal time frames of two years, consisting of a pre-pandemic, pandemic, and post-pandemic period, in which preoperative and perioperative as well as postoperative parameters were compared. **Results**: Statistically significant differences regarding environment, type of admission, and ASA score, as well as a more advanced tumoral stage, increased number of important postoperative complications, and a lower minimally invasive surgical approach, were highlighted within the pandemic group. Statistically significant differences regarding emergency diagnosis as well as late diagnosis were highlighted. There were no significant differences regarding the tumor location, postoperative 30-day mortality, or hospitalization duration. **Conclusions**: COVID-19 significantly impacted the surgical timing in colorectal cancer, as well as addressability for the rural population, with a marked decrease in elective cases as well as an increased number of cases diagnosed in an emergency setting, with locally advanced tumors. However, no significant changes in postoperative mortality or hospitalization duration were highlighted. In addition, most of the changes highlighted were reverted in the post-pandemic period. Further studies are required to observe the long-term effects in terms of morbidity and mortality, regarding the delay of diagnosis and oncological treatment.

## 1. Introduction

Severe acute respiratory syndrome coronavirus 2 (SARS-CoV-2) is a coronavirus responsible for the global COVID-19 pandemic, which began in January 2020 and was first identified in Wuhan, China, in December 2019. It belongs to the *Coronaviridae* family, which includes other well-known viruses such as SARS-CoV and MERS-CoV, responsible for previous outbreaks that caused severe respiratory illnesses [1,2]. Spreading through droplets and aerosols, as well as via contaminated surfaces, it is particularly notable for its ability to infect a broad range of cells, most prominently those of the respiratory tract. This can result in a wide array of symptoms, ranging from mild respiratory infections to severe, life-threatening complications [3]. Furthermore, COVID-19, which remains an ongoing pandemic since late 2019, is considered one of the deadliest pandemics, with its impact being especially severe in patients with pre-existing comorbidities [1,2,3].

One of the primary effects COVID-19 had on the healthcare system was the reallocation of healthcare resources [4,5]. With the focus primarily on containing the spread of the virus and providing care for severe cases, individuals with other chronic conditions sought less medical care. This was either to avoid increasing the burden on the healthcare sector or due to the heightened risk of exposure in healthcare facilities [6,7]. As a result, the number of emergency cases and follow-up visits for chronic conditions decreased significantly, affecting outpatient clinics and ambulatory services worldwide [8].

Surgical departments were among the most impacted sectors of healthcare during the COVID-19 pandemic [9,10]. Access to both elective and emergency surgeries was significantly reduced, with various tertiary centers reporting a decrease of up to 40% in new cases and elective surgeries [11,12]. Moreover, elective surgeries for newly diagnosed cancer showed a declining trend during the pandemic, particularly for minimally invasive and elective procedures. There was a shift toward conservative treatments or approaches requiring shorter hospitalization durations to minimize patient exposure risk [13]. Furthermore, some studies highlighted significant delays in surgeries to reduce patient exposure. Although there were minimal differences in postoperative outcomes, further research is necessary to determine whether these delays impacted overall patient survival, particularly among oncological patients, during the pandemic [14,15].

Colorectal cancer (CRC) is no exception to the changes in treatment observed during the COVID-19 pandemic, with most Western countries reporting a decrease in the number of CRC surgeries in 2020. Additionally, an increased number of cases presenting with locally advanced CRC were observed in subsequent years (2021–2023) [14,16]. Studies have also indicated a preference for open surgery over minimally invasive surgery during the pandemic due to concerns about aerosol spread of the coronavirus during desufflation. However, these concerns were later dismissed as the risk–benefit analysis favored laparoscopic surgery [9,17].

Alternative therapeutic strategies, such as stent placement for obstructive colon cancer and the introduction of new neoadjuvant therapies, were proposed to manage the caseload and minimize exposure risk. These measures resulted in increased endoscopic and palliative care caseloads [18]. The implementation of these strategies varied between countries, depending on the quality of their healthcare systems and the status of the pandemic [19]. Moreover, discrepancies in lockdown durations and restrictive hospital measures between developed and developing countries may have influenced outcomes. Therefore, a single-center study approach might be more appropriate to account for these differences.

The purpose of this retrospective, single-center study is to evaluate the short-term and post-pandemic effects on the surgical management of CRC. Specifically, it aims to determine whether these changes led to a shift in case severity at presentation and whether they impacted overall patient survival. Additionally, a comparison was conducted between COVID-negative and COVID-positive patients during the pandemic, focusing on postoperative complications, 30-day mortality rates, and hospitalization duration to reduce bias.

## 2. Materials and Methods

This retrospective comparative study analyzed data from a single tertiary center (Surgical Clinic No. 3, Regional Institute of Gastroenterology and Hepatology “Octavian Fodor,” Cluj-Napoca) over six years. Data were collected from the institution’s database and categorized into three groups: Group A—pre-pandemic (15 March 2018–15 March 2020), Group B—pandemic (15 March 2020–15 March 2022), and Group C—post-pandemic (15 March 2022–15 March 2024). Timeframes were selected based on the duration of pandemic restrictions at the tertiary center, with equal intervals chosen for all three groups.

The study adhered to the Strengthening the Reporting of Observational Studies in Epidemiology (STROBE) Statement, ensuring compliance with all major criteria. The primary criterion for group division was the implementation and subsequent lifting of COVID-19 restrictions. These restrictions began on 15 March 2020, and ended on 15 March 2022. Each group was designed to encompass an equal time span.

All patients in Group B underwent mandatory COVID-19 testing. In contrast, testing in Group C was limited to patients exhibiting flu-like symptoms.

The primary inclusion criteria were as follows:

(a) Patients who signed informed consent for data collection.

(b) Patients over the age of 18 with colorectal cancer (CRC) who underwent either elective or emergency surgery.

The primary exclusion criteria were as follows:

(a) Patients who did not provide informed consent for data collection.

(b) Patients diagnosed with CRC who did not undergo resection (e.g., cases involving exploratory surgery or simple colostomy).

Additionally, patients with negative biopsy results were excluded from the study. Collected data were compared among the three groups. Patients from Group B were further subdivided into COVID-negative and COVID-positive cohorts, with perioperative and postoperative data analyzed for each subgroup. A comprehensive analysis of eligible patients’ medical records was conducted, dividing the data into preoperative, perioperative, and postoperative categories.

Preoperative data: Age, BMI, gender, symptoms at presentation, the type of admission, environment, geographic location, and ASA score were recorded. Oncological data, including tumor location, stage, grade, histology, and the presence of metastases, were also collected.

Perioperative data: The type of surgery (curative or palliative), presence of intraoperative complications, and surgical approach were analyzed.

Postoperative data: Postoperative complications, hospitalization duration, and 30-day mortality were evaluated.

A specific comparison of postoperative complications was performed between the COVID-negative and COVID-positive patients within Group B.

Regarding geographic location, patients were divided into two groups:

Patients residing in the Transylvanian Region, where the tertiary center is located.

Patients from other regions (Wallachia, Moldova, or foreign patients).

Data were collected using the Microsoft Excel 2021 program and were interpreted through the IBM SPSS v26.0 software (Chicago, IL, USA), using parametric as well as non-parametric tests based on the distribution of data. Distribution was verified using Kolmogorov Smirnov and Shapiro–Wilk tests. Categorical variables were interpreted using the Pearson Chi-Squared test. Correlation between numerical variables was highlighted using the Pearson test. A comparison between groups of continuous variables was performed using the ANOVA test for normally distributed data and Kruskal–Wallis test for non-normally distributed data between three or more groups. For comparison between two groups, a T-test for independent variables was performed for normally distributed data, and a Mann–Whitney U test was performed for non-normally distributed data. Disease-free survival at 12 months was also evaluated, using log rank tests. Confidence intervals were at a 95% level and the threshold of statistical significance was *p* < 0.05.

To conduct the present research, we employed the following endpoints: the primary endpoint was whether the pandemic restrictions themselves influenced the diagnosis as well as the presentation of colorectal cancer cases, and the secondary endpoints were regarding COVID-19 infection as well as an overview of the post-pandemic colorectal cancer caseload in our center, most specifically, whether the post-lockdown changes would revert the caseload outcome to the previous values.

## 3. Results

A total of 1687 patients were included in the study. A total of 686 cases underwent surgery in the pre-pandemic period (group A), while 411 patients underwent surgery during the pandemic period (group B) and 590 patients underwent surgery in the post-pandemic period. All patients during the pandemic period were tested for COVID-19 at admission, as well as during the admission if they presented specific symptoms. Group C did not require mandatory testing; however, patients exhibiting COVID-19 symptoms were tested with no specific influence on their cohort. Preoperative data are highlighted in Table 1.

Based on the type of admission, statistically significant differences were highlighted, with a higher level of emergency cases in group B (29.4% vs. 44.8% vs. 32.1%, *p* = 0.001). In addition, a statistically significant difference of ASA scores was highlighted between the three groups, with a higher prevalence of ASA 4 cases being present in group B (2.6% vs. 12.6% vs. 2.4%, *p* = 0.001). Additional statistically significant differences were highlighted regarding environment (rural—41.6% vs. 29.2% vs. 46.3%, *p* = 0.001), as well as age, with the highest mean age being present in group B (64.9 years, *p* = 0.04). No statistically significant differences regarding gender or mean BMI were highlighted in our study.

When accounting for symptoms at presentation, the most frequent symptoms found were changes in bowel habits as well as malignancy-specific symptoms. Significant differences between the groups were found in alarm symptoms, most notably abdominal pain (29.3% vs. 47.9% vs. 32.2%, *p* = 0.004), as well as anemia (17.2% vs. 29.4% vs. 19.1%, *p* = 0.01) and intestinal obstruction (19.2% vs. 27.0% vs. 23.3%, *p* = 0.02), which are suggestive of an emergency presentation. Detailed percentage values are highlighted in Table 2.

In terms of surgical approach, there was a statistically significant decrease in minimally invasive treated cases in group B (30.3% vs. 18.4% vs. 31.8%, *p* = 0.001), without any major differences between group A and C. In addition, there were statistically significant differences regarding the CRC stage at presentation, with a higher percentage of advanced cases being prevalent in group B (42.0% vs. 52.4% vs. 46.2%, *p* = 0.003). No statistically significant differences regarding the tumor location between the three groups have been observed, the most frequent tumors overall being the left colon tumors. Perioperative parameters are highlighted in Table 3.

There were no statistically significant differences regarding hospitalization duration (6.14 vs. 6.43 vs. 6.01, *p* = 0.34); however, there was a statistically significant increase in stoma formation in group B (14.7% vs. 21.4% vs. 16.6%, *p* = 0.01), as well as a significant increase in the postoperative complications higher than grade III (4.3% vs. 9.2% vs. 4.7%). However, no statistically significant differences regarding the 30-day mortality rate were found (1.6% vs. 3.6% vs. 2.0%, *p* = 0.07). Furthermore, we analyzed the 12-month disease-free survival between the two groups, and no statistically significant differences were found (91.6% vs. 93.4% vs. 89.4%). These results are highlighted in Table 4.

When accounting for patients who became COVID positive during admission or at admission within group B, there was a statistically significant difference regarding the postoperative complication rate, compared with COVID-negative patients (3.4% vs. 18.4, *p* = 0.001), as well as the 30-day mortality rate (1.5% vs. 10.8%, *p* = 0.001). There were no statistically significant differences regarding mean hospitalization duration as well as stoma formation between the two groups. These changes are highlighted in Table 5.

When taking into consideration the geographic location of the patients, there were statistically significant differences between the three groups, with a higher number of patients coming straight from the Transylvanian Region being diagnosed during the pandemic period than in the other two periods (74.9% vs. 87.1% vs. 76.6%, *p* = 0.02). In addition, more than 75% of the patients were diagnosed in the same center during the COVID-19 lockdown, with the difference being statistically significant when compared with the other groups. These differences are highlighted in Table 6.

## 4. Discussion

The COVID-19 pandemic significantly impacted the timing of surgeries and access to adequate oncological treatment worldwide. Multiple studies have highlighted delays in surgical care for oncological patients due to COVID-19 infections and full lockdowns in certain countries [20,21]. Some studies reported a substantial reduction—up to 35%—in the number of oncological surgeries, accompanied by a marked increase in postoperative morbidity and mortality rates [22,23,24].

Our study focused primarily on the COVID-19 restriction period compared to the pre-pandemic period, with additional emphasis on the post-pandemic period to assess potential improvements in colorectal cancer (CRC) surgical care. We observed a 40% decrease in CRC surgical treatments during the lockdown period. However, the post-pandemic period demonstrated recovery, with surgical numbers comparable to those from the pre-pandemic period.

These findings align with most studies conducted at tertiary centers [20,21,23], but the results may vary between countries due to differences in lockdown durations and the specific measures and laws implemented during the pandemic. In more developed countries, where lockdowns were shorter, postoperative outcomes were minimally affected. However, a diagnostic delay was frequently observed, although its influence on overall survival has yet to be fully determined [20,23].

Our study also revealed a marked percentage increase in admissions requiring urgent care. Despite the absolute number of admissions remaining relatively unchanged, this underscores the impact of early lockdown restrictions, reduced healthcare accessibility during the pandemic, and delayed surgical interventions for colorectal cancer (CRC) patients. Consequently, there was a significantly lower number of elective cases treated during this period. These findings align with several studies, including those conducted at tertiary centers, multicenter analyses, and global studies [25,26,27]. Moreover, we highlighted no statistically significant differences regarding the 12-month disease free survival rate between the groups. Another angle for our study would be highlighting the 3-year DFS and 5-year DFS, as well as the 5-year overall survival, further data requiring centralization.

It has been hypothesized that patients may have delayed seeking medical attention due to fear of contracting the coronavirus at the height of the pandemic or to avoid overburdening the healthcare system. Furthermore, our study demonstrated a significantly higher proportion of advanced-stage CRC cases during the lockdown period. This supports arguments regarding challenges in resource allocation and the predominance of urgent care cases over elective ones.

Similar findings have been reported in studies involving other oncological case series. Additionally, our study highlighted elevated rates of “alarm symptoms” among patients presenting during the lockdown, many of which typically necessitate emergency care. These observations further emphasize the effects of pandemic-related disruptions on timely cancer diagnosis and treatment.

The statistically significant changes in demographic characteristics observed in our study can be attributed to the limited access to healthcare for rural populations during the lockdown and challenges in the adequate distribution of emergency services [28,29]. Although this study focuses solely on colorectal cancer (CRC), similar studies have highlighted the difficulty of allocating resources to rural areas during the pandemic. Additionally, the urban location of most tertiary centers posed further challenges for rural patients seeking care, due to pandemic-related restrictions and the strain on emergency transportation systems.

While the observed age differences between the three groups were statistically significant, the gap was relatively small. This may be explained by a higher prevalence of older patients presenting to emergency departments, likely due to a higher frailty index or multiple comorbidities that necessitated urgent care. However, further studies are needed to substantiate this hypothesis.

Furthermore, our study exhibited a significant increase in cases diagnosed in the same center during the pandemic period, compared to the other two periods. This may as well be attributed to the lockdown restrictions as well as the low addressability for the outpatient clinic, during the COVID period, especially from other regions and centers.

One of the most important highlights of our study remains the significant decrease in minimally invasive surgeries. This can be attributed to two major factors: the primary factor represented by the delayed surgical timing which may offset the diagnosis and the prognosis of the patient [30,31,32]. Patients present either in an emergency setting or in an elective setting with a locally advanced tumor that might not be able to be removed through a minimally invasive approach. Despite this statement being supported by some studies, it still requires more multicentric studies and it is very difficult to pinpoint the treatment delay. However, some studies pinpoint that these effects may be long-term, not only in colorectal cancer but in other areas of oncology which in time may increase morbidity due to oncological diagnosis and treatment delay because of the pandemic [31,32]. The other major factor remains the potential risk of laparoscopy in potentially positive patients. Since laparoscopy is considered an aerosol-forming procedure, some studies have shown that it may have a potential risk of viral transmission, especially during desufflation, thus prompting surgeons to change their tactics [33]. As the pandemic progressed, this paradigm gradually shifted, and with better control of the outbreak, as well as enforcing more strict rules, the benefits of laparoscopy outweighed the small risk of viral transmission; thus, by the end of the lockdown, more and more studies emerged focusing on these aspects [34,35]. This was, however, applicable only in the tertiary centers with better lockdown measures, higher quality infrastructure, and state-of-the-art equipment, although more studies are required to highlight this aspect.

Another important highlight was the increased number of stomas which were performed during the pandemic period. This can also be associated with a higher ASA score, and thus, patients presenting a potentially higher risk of anastomotic leak, therefore reducing the risk of reinterventions [31,36,37]. These results are supported by other similar studies. Moreover, there was an increased number of high-level Clavien-Dindo postoperative complications during the pandemic. When looking at the data analyzed based on COVID-19 infection status, these parameters are significantly increased in COVID-positive patients [9,37,38,39]. Having an elevated proinflammatory status, as well as an increased frailty index, can lead to certain postoperative complications, as highlighted by other studies [40,41,42]. It is worth noting, however, that there were little to no differences regarding the hospitalization duration. Having well-established protocols as well as taking precautions such as those mentioned above, may have minimized the hospitalization duration to minimize the risk of COVID-19 exposure, especially in oncological patients.

Despite not being statistically significant, there was a marked increase regarding 30-day mortality during the pandemic period, which is especially highlighted in the COVID-positive patients. However, taking the COVID-positive patients out of consideration, the levels of mortality remain the same between the three groups. This statement is further supported by other studies [43,44,45]. In addition, some tertiary centers highlighted lower levels of mortality, mainly due to lower postoperative complication rates, which were attributed to better case selection as well as more stringent protocols [44,45]. Having a lower caseload of oncological surgeries, with carefully selected patients, led to overall better postoperative outcomes and overall better short-term survival. However, the long-term survival effects are yet to be determined, which might be the basis for further studies.

This study has some limitations, primarily due to the retrospective approach of the analysis, as well as possible data collection bias. However, these limitations are minimized based on the large sample size as well as addressing the COVID-positive patient bias. Furthermore, having a level of comparison before as well as after the pandemic lockdown period may minimize the bias, thus strengthening the significance of the results obtained. Another important limitation is that the database contains patients from a single-center study. Despite accounting for a high number of patients with CRC, over 80% of cases being from the entire Transylvanian region, the data we found should be compared with other centers, not only from the Transylvania region, but from the other two Romanian regions as well, especially since they share the same healthcare system, and mostly, they present similar demographic groups. Although our study might be more relevant for the Transylvanian region, we are unable to assess its strength compared to other similar studies from this region. A regional or nation-wide study, or the enrollment of this data into a multi-center study, would strengthen the results. In addition, further oncological pathologies can be addressed through this method. This, however, can be considered as an advantage of this study’s results, which focus on a single pathology, thus further limiting the risk of case selection bias. Finally, one of the more important aspects is represented by the lockdown periods and variability in restrictions. Based on certain aspects of the pandemic in Romania, more stringent restrictions or fewer restrictions were imposed which created a bias of variability during the two-year lockdown period. Therefore, in the early months of the pandemic, the restrictions were more severe, and thus, the number of cases was lower. As the pandemic progressed, some of the restrictions were lifted; however, due to variability in overall COVID-19 cases, these restrictions might have been re-imposed, and thus it is very difficult to pinpoint certain lockdown periods during our two-year cohort. This might be considered a limitation in our study.

## 5. Conclusions

COVID-19 significantly impacted the surgical timing in colorectal cancer surgery, as well as addressability, especially for the rural population. Despite tightening rules and maintaining strict protocols, there was a significant increase in important postoperative complications, with a significant decrease in the usage of minimally invasive surgery. However, no significant changes regarding postoperative mortality or hospitalization duration were highlighted. In addition, most of the changes highlighted during the COVID-19 pandemic were reverted after the lockdown period, in terms of emergency caseload, symptoms at presentation, and diagnosis-to-surgery timing. Further studies are required to observe the long-term effects of delaying the surgical timing in terms of morbidity and mortality.

## Figures and Tables

**Table 1 diagnostics-15-00129-t001:** Preoperative parameters.

Parameters		Group A (n = 686)	Group B (n = 411)	Group C (n = 590)	*p* Value
Mean age (years)		63.45 (±11.34)	64.92 (±9.72)	63.28 (±10.97)	**0.04**
Gender	Male	394 (57.4%)	248 (60.3%)	342 (57.9%)	0.62
	Female	292 (42.6%)	163 (39.7%)	248 (42.1%)	
Environment	Urban	401 (58.4%)	291 (70.8%)	317 (53.7%)	**0.001**
	Rural	285 (41.6%)	120 (29.2%)	273 (46.3%)	
Mean BMI		24.91 (±4.19)	25.02 (± 4.55)	25.21 (± 4.72)	0.48
Type of admission	Elective	491 (71.6%)	227 (55.2%)	401 (67.9%)	**0.001**
	Emergency	195 (29.4%)	184 (44.8%)	189 (32.1%)	
ASA score	1	83 (12.1%)	14 (3.4%)	72 (12.2%)	**0.001**
	2	298 (43.4%)	148 (36.0%)	259 (43.8%)	
	3	287 (41.8%)	197 (47.9%)	245 (41.6%)	
	4	18 (2.6%)	52 (12.6%)	14 (2.4%)	

**Table 2 diagnostics-15-00129-t002:** Symptoms at presentation.

Symptoms at Presentation	Group A (n = 686)	Group B (n = 411)	Group C (n = 590)	*p* Value
Weight loss	390 (56.8%)	204 (49.6%)	298 (50.5%)	0.19
Fatigue	321 (46.7%)	198 (48.2%)	243 (41.1%)	0.38
Change in bowel habits	501 (73.1%)	309 (75.1%)	437 (74.0%)	0.72
Abdominal pain	201 (29.3%)	197 (47.9%)	194 (32.2%)	0.004
Anemia	118 (17.2%)	121 (29.4%)	113 (19.1%)	0.01
Intestinal obstruction	132 (19.2%)	111 (27.0%)	138 (23.3%)	0.02

**Table 3 diagnostics-15-00129-t003:** Perioperative parameters and distribution regarding surgical approach.

Parameters		Group A (n = 686)	Group B (n = 411)	Group C (n = 590)	*p* Value
Tumor location	Right colon	194 (28.3%)	105 (30.0%)	155 (26.2%)	0.8
	Transverse colon	101 (14.7%)	64 (15.5%)	104 (17.7%)	
	Left colon	221 (32.2%)	140 (34.0%)	191 (32.4%)	
	Rectum	170 (24.7%)	102 (24.8%)	140 (23.7%)	
Stage	I or II	398 (58.0%)	196 (47.6%)	318 (53.8%)	**0.003**
	III+	288 (42.0%)	215 (52.4%)	272 (46.2%)	
Surgical approach	Open	478 (69.7%)	335 (81.6%)	402 (68.2%)	**0.001**
	Minimally invasive	208 (30.3%)	76 (18.4%)	188 (31.8%)	

**Table 4 diagnostics-15-00129-t004:** Postoperative parameters, complications, and mortality rate.

Parameters	Group A (n = 686)	Group B (n = 411)	Group C (n = 590)	*p* Value
Stoma Formation	101 (14.7%)	88 (21.4%)	98 (16.6%)	**0.01**
Mean Hospitalization Duration	6.14 (± 0.84)	6.43 (± 0.91)	6.01 (± 0.81)	0.34
Clavien Dindo III+	30 (4.3%)	28 (9.2%)	28 (4.7%)	**0.001**
30-day mortality	11 (1.6%)	15 (3.6%)	12 (2.0%)	0.07
12-month disease free survival	629 (91.6%)	384 (93.4%)	528 (89.4%)	0.38

**Table 5 diagnostics-15-00129-t005:** Postoperative parameters, complications, and mortality rate based on COVID status in Group B.

Parameters	COVID Negative (n = 319)	COVID Positive (n = 92)	*p* Value
Clavien-Dindo III+	11 (3.4%)	17 (18.4%)	**0.001**
Stoma formation	63 (19.7%)	25 (27.1%)	0.08
30-day mortality	5 (1.5%)	10 (10.8%)	**0.001**
Mean hospitalization	6.03 (± 0.81)	8.08 (± 2.34)	0.1

**Table 6 diagnostics-15-00129-t006:** Geographic parameters of the patients.

Parameters		Group A (n = 686)	Group B (n = 411)	Group C (n = 590)	*p* Value
Geographic location	Other regions	172 (25.1%)	53 (12.9%)	138 (23.4%)	0.02
	Transylvania	514 (74.9%)	358 (87.1%)	452 (76.6%)	
Diagnosed in the same center	Yes	390 (56.8%)	309 (75.1%)	408 (69.2%)	0.001
	No	296 (43.2%)	102 (24.9%)	182 (30.8%)	

## Data Availability

The authors confirm that the data supporting the findings of this study are available within the article.

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
