# Peer review of "Influence of SARS-CoV2 Pandemic on Colorectal Cancer Diagnosis, Presentation, and Surgical Management in a Tertiary Center: A Retrospective Study"

_diagnostics, 2025, doi:10.3390/diagnostics15020129_

Round 1

Reviewer 1 Report

Comments and Suggestions for Authors

1. The author does not offer any significant new insights into the effects of the COVID-19 pandemic on colorectal cancer diagnosis and management. The findings largely confirm well-established trends reported in the literature without presenting any novel hypotheses, techniques, or analytical frameworks.

2. The study’s methodology is insufficiently detailed. There is a lack of clarity regarding how patients were selected for each group (pre-pandemic, pandemic, post-pandemic), and potential biases in patient selection have not been addressed. This undermines the validity of the comparative analysis.

3. While some statistical significance is reported, there is no detailed description of the statistical methods used. Key analyses, such as multivariate regression or survival analysis, are absent, which limits the ability to assess the relative contribution of different factors to the observed outcomes.

4. This study does not adequately address confounding factors such as variations in healthcare resource allocation, changes in screening programs, or differences in patient comorbidities during the pandemic.

5. As a single-center study, the findings are limited in their applicability to broader populations. The authors do not discuss whether the results can be generalized to other regions, healthcare systems, or demographic groups, reducing the overall impact of the study.

6. The discussion section does not critically evaluate the findings in the context of existing literature. For example, there is no exploration of why certain parameters, such as tumor stage or minimally invasive surgery rates, were affected, while others, like postoperative mortality, were not.

7. Although the conclusion mentions the need for further studies on long-term morbidity and mortality, the current study does not provide any meaningful preliminary data or insights in this regard.

Comments on the Quality of English Language

The English is understandable but requires grammatical corrections, improved sentence structure, and more precise technical terminology for clarity and professionalism.

Author Response

Greetings, and thank you for offering a couple of insights as well as some suggestions for our article. Your feedback is greatly appreciated, therefore we decided to answer your suggestions in a point-by-point.

1. The author does not offer any significant new insights into the effects of the COVID-19 pandemic on colorectal cancer diagnosis and management. The findings largely confirm well-established trends reported in the literature without presenting any novel hypotheses, techniques, or analytical frameworks.

4. This study does not adequately address confounding factors such as variations in healthcare resource allocation, changes in screening programs, or differences in patient comorbidities during the pandemic.

I think these two questions can be addressed here. First, the current experience is highlighted through the eyes of a tertiary center, which is responsible for a good part of the colorectal cases in Transylvania. More than 80% of the cases are from the Cluj county as well as the neighboring counties which are encompassed in the Transylvanian region. Despite not being the only center from Transylvania which deals with colo-rectal cancer, it can show some insight regarding the way our community dealt with the coronavirus restrictions during that timeframe. A much more interesting insight would be expanding it to a multicentric study, which might be a point for further studies, for which we're grateful for your suggestion. Second, we agree to the extent that we didn't dive in the comorbidity chapter for COVID-19. These paragraphs were further expanded in the Material and Methods as well as in the discussion part, and we thank you for the suggestions.

2. The study’s methodology is insufficiently detailed. There is a lack of clarity regarding how patients were selected for each group (pre-pandemic, pandemic, post-pandemic), and potential biases in patient selection have not been addressed. This undermines the validity of the comparative analysis.

This was one of the suggestions highlighted by all of the reviewers, therefore we completely revised it. However the potential biases were addressed in the discussion section. We decided to further expand the paragraph regarding the potential biases, and included it in the limits of our study.

3. While some statistical significance is reported, there is no detailed description of the statistical methods used. Key analyses, such as multivariate regression or survival analysis, are absent, which limits the ability to assess the relative contribution of different factors to the observed outcomes.

Although the end-point wasn't to highlight the overall survival of the patients in our study, we went with your suggestion to improve our data analysis by implementing a multivariate analysis regarding the potential factors impacted by COVID-19. Furthermore, we've detailed the material and methods paragraph describing the current statistical methods used. This is the primary reason for why our response took us so long however we're again grateful for the suggestion.

5. As a single-center study, the findings are limited in their applicability to broader populations. The authors do not discuss whether the results can be generalized to other regions, healthcare systems, or demographic groups, reducing the overall impact of the study.

As stated previously (see 1 and 4), we've expanded this point in the discussions section. Although it may be difficult to pinpoint on certain demographic groups, these results can be generalized to the other two regions of Romania (Moldova and Wallachia), as they share similarities regarding healthcare, university centres, as well as population. We've managed to pinpoint these aspects in the discussions section.

6. The discussion section does not critically evaluate the findings in the context of existing literature. For example, there is no exploration of why certain parameters, such as tumor stage or minimally invasive surgery rates, were affected, while others, like postoperative mortality, were not.

We've overlooked on our collected data as well as into the current literature, in order to expand this point. As we mentioned, one of the explanations for lower MIS rates would be the potential risk of airborne spread of the virus during dessuflation, which was supported by some studies in the early COVID-19 lockdown stage (we highlighted this in the discussion section). All of the parameters mentioned above were highlighted in the discussion section.

7. Although the conclusion mentions the need for further studies on long-term morbidity and mortality, the current study does not provide any meaningful preliminary data or insights in this regard.

We've updated the paragraph regarding long-term morbidity as well as mortality, as indeed we did not delve too much into this aspect, and we apologize for that.

Furthermore, we did an extensive spell-check on our article, and we trimmed most of the bloated paragraphs as well as any redundant spelling. If any other suggestions might be to your consideration, we would be more than welcome to hear them. Thank you for taking your time in reviewing our article.

Reviewer 2 Report

Comments and Suggestions for Authors

This interesting article highlights the difficulties in the management of colorectal cancer during the pandemic era. Although a substantial number have been reported about this era, the topic continues to raise our interesting. Besides, our memories are still fresh. 

The presentation is quite understandable. However, there are long sentences that need shortness. That means a revision of them by an experienced native author.

The method meets the standard requirements

The statistical analysis is appropriate and the results are presented clearly.

The discussion covers the spectrum of the main issues.

The conclusions are compatible with the main results

Author Response

"

This interesting article highlights the difficulties in the management of colorectal cancer during the pandemic era. Although a substantial number have been reported about this era, the topic continues to raise our interesting. Besides, our memories are still fresh. 

The presentation is quite understandable. However, there are long sentences that need shortness. That means a revision of them by an experienced native author.

The method meets the standard requirements

The statistical analysis is appropriate and the results are presented clearly.

The discussion covers the spectrum of the main issues.

The conclusions are compatible with the main results"

We thank you for the through review and the kind words. One of the primary issues regarding English was due to the incongruence between the written paragraphs, since there were more authors that contributed to it. We've managed to input your changes, as well as rephrase some of the parts. In addition we've added further changes to the article per other reviewers' suggestions.

Reviewer 3 Report

Comments and Suggestions for Authors

The authors present a retrospective research concerning the colon cancer surgery during Covid pandemic and its comparison to the pre and post pandemic era.

It has to be emphasized that the manuscript is retrospective, therefore, how is it possible that the one of the inclusion criteria was the informed consent from the patients. Does all the patients during admission signed blank informed consent for data gathering? Doesn’t it produce ethical concerns in the author’s country? Why did the author’s excluded and included patients who benefited from surgery? What does it mean that the patient benefited from surgery? Does it mean that the patients survived? Was discharged? Was disease free in the follow up period? Did you test the patients for Covid in group C? End of pandemic doesn’t mean that there are no patients suffering from this disease. To conclude, the material and method section has to be improved.

The results, discussion sections are well presented and is very interesting.

Author Response

Greetings, and thank you for taking your time to review our manuscript. We're going to answer to your suggestions point-by-point.

"It has to be emphasized that the manuscript is retrospective, therefore, how is it possible that the one of the inclusion criteria was the informed consent from the patients. Does all the patients during admission signed blank informed consent for data gathering? Doesn’t it produce ethical concerns in the author’s country?"

It is emphasized in the Material and Methods section. Since it's a University Hospital, most of the informed written consents have a "data collection" addendum, in which their medical data is stored anonymously for database development with its primary focus is for research. We've rephrased this part, as it can cause confusion. Patients did not sign an informed consent for a retrospective study, they signed the standard consent form for admission as well as a consent form for anonymous data collection (which is mandatory in any hospital, especially in a Uni hospital). Thank you for pin-pointing that out, and we're sorry for the confusion.

Why did the author’s excluded and included patients who benefited from surgery? What does it mean that the patient benefited from surgery? Does it mean that the patients survived? Was discharged? Was disease free in the follow up period?

Since it was a retrospective analysis, we did not want to focus on minimal surgery (only stoma formation or exploratory laparoscopy/laparotomy). We intended to focus on the patients who underwent tumoral resection. We've rephrased this part in the Material and Methods section, and we apologize for the confusion. We did not focus on the disease free part, as it was not the main endpoint of this study, but this could be another endpoint for further studies from our center. As far as we looked in the literature, there are studies that do not highlight a decrease in DFS. We thank you for pointing that out, though. 

Did you test the patients for Covid in group C? End of pandemic doesn’t mean that there are no patients suffering from this disease. To conclude, the material and method section has to be improved."
We did perform tests for the patients in group C, but we have specifically selected this group, as mandatory testing was no longer required after that date. Some of the patients from group C underwent testing due to the presence of flu-like symptoms but they weren't attributed to COVID-19. Mostly, the patients who underwent elective surgery during late 2022 and early 2023 were informed to notify us for flu-like symptoms and if they indeed exhibited specific symptoms they were advised to postpone their surgery. We've included this at the Material and Methods as well as the discussion sections.

To summarize, we thank you for highlighting the gaps and eventual hiccups in explaining the case-selection. We've improved the Material and Methods as well as the discussion section suggested by you as well as the other reviewers. We hope that this answers your questions. Thank you for helping us improve our study.

Round 2

Reviewer 1 Report

Comments and Suggestions for Authors

The authors did not adequately address most of the comments, and I am not satisfied with the corrections in this revised version. The paper still suffers from significant weaknesses, including:

1. The study fails to provide any meaningful new insights into the effects of the COVID-19 pandemic on colorectal cancer diagnosis and management. While the authors claim to highlight the regional perspective, the findings remain redundant, merely confirming already well-documented trends in the literature. No innovative hypotheses, analytical frameworks, or theoretical contributions have been presented, which significantly diminishes the value of the study.

2. Despite revisions, the methodology remains vague and lacks critical details. The authors do not clearly explain the patient selection process for the pre-pandemic, pandemic, and post-pandemic groups, which introduces selection bias and threatens the study's internal validity. Additionally, while the authors mention multivariate analysis, there is still no evidence of rigorous statistical methods such as survival analysis or proper handling of confounding factors. The reported statistical results remain superficial and insufficient to draw valid conclusions.

3. As a single-center study, the findings lack generalizability to broader populations. The authors’ attempt to argue for regional applicability is unsupported by comparative data or evidence. Without a multicenter approach or diverse demographic representation, the study’s findings cannot be extrapolated to other healthcare systems or regions, undermining its external validity.

Comments on the Quality of English Language

The quality of the English language in this manuscript still requires improvement. While some attempts have been made to address previous suggestions, the text continues to suffer from grammatical errors, awkward phrasing, and inconsistent sentence structure.

Reviewer 3 Report

Comments and Suggestions for Authors

The manuscript has been improved and my concerns have been dispeled